# Diagnosis of Fungal Keratitis in Low-Income Countries: Evaluation of Smear Microscopy, Culture, and In Vivo Confocal Microscopy in Nepal

**DOI:** 10.3390/jof8090955

**Published:** 2022-09-13

**Authors:** Jeremy J. Hoffman, Reena Yadav, Sandip Das Sanyam, Pankaj Chaudhary, Abhishek Roshan, Sanjay Kumar Singh, Simon Arunga, Victor H. Hu, David Macleod, Astrid Leck, Matthew J. Burton

**Affiliations:** 1International Centre for Eye Health, London School of Hygiene and Tropical Medicine, London WC1E 7HT, UK; 2Sagarmatha Choudhary Eye Hospital, Lahan 56500, Nepal; 3Department of Ophthalmology, Mbarara University of Science and Technology, Mbarara P.O. Box 1410, Uganda; 4MRC International Statistics & Epidemiology Group, London School of Hygiene & Tropical Medicine, London WC1E 7HT, UK; 5National Institute for Health Research Biomedical Research Centre for Ophthalmology at Moorfields Eye Hospital NHS Foundation Trust and UCL Institute of Ophthalmology, London EC1V 9EL, UK

**Keywords:** microbial keratitis, fungal keratitis, in vivo confocal microscopy, diagnosis, microbiology, Nepal, cornea, culture, microscopy

## Abstract

Clinically diagnosing fungal keratitis (FK) is challenging; diagnosis can be assisted by investigations including in vivo confocal microscopy (IVCM), smear microscopy, and culture. The aim of this study was to estimate the sensitivity in detecting fungal keratitis (FK) using IVCM, smear microscopy, and culture in a setting with a high prevalence of FK. In this cross-sectional study nested within a prospective cohort study, consecutive microbial keratitis (MK) patients attending a tertiary-referral eye hospital in south-eastern Nepal between June 2019 and November 2020 were recruited. IVCM and corneal scrapes for smear microscopy and culture were performed using a standardised protocol. Smear microscopy was performed using potassium hydroxide (KOH), Gram stain, and calcofluor white. The primary outcomes were sensitivities with 95% confidence intervals [95% CI] for IVCM, smear microscopy and culture, and for each different microscopy stain independently, to detect FK compared to a composite referent. We enrolled 642 patients with MK; 468/642 (72.9%) were filamentous FK, 32/642 (5.0%) were bacterial keratitis and 64/642 (10.0%) were mixed bacterial-filamentous FK, with one yeast infection (0.16%). No organism was identified in 77/642 (12.0%). Smear microscopy had the highest sensitivity (90.7% [87.9–93.1%]), followed by IVCM (89.8% [86.9–92.3%]) and culture (75.7% [71.8–79.3%]). Of the three smear microscopy stains, KOH had the highest sensitivity (85.3% [81.9–88.4%]), followed by Gram stain (83.2% [79.7–86.4%]) and calcofluor white (79.1% [75.4–82.5%]). Smear microscopy and IVCM were the most sensitive tools for identifying FK in our cohort. In low-resource settings we recommend clinicians perform corneal scrapes for microscopy using KOH and Gram staining. Culture remains an important tool to diagnose bacterial infection, identify causative fungi and enable antimicrobial susceptibility testing.

## 1. Introduction

Microbial keratitis (MK) is an ocular emergency; without prompt, appropriate treatment, significant ocular morbidity can ensue, including blindness through corneal scarring and even eye loss [1]. As there is a diverse range of causative organisms—bacteria, fungi, protozoa and viruses—correct treatment depends on accurately identifying the microbe responsible through clinical examination and suitable investigations [2]. In tropical low- and middle-income countries, fungal infection can account for more than half of MK cases [3], meaning a key consideration for management is whether the organism is bacterial or fungal. Unfortunately, making a diagnosis on clinical grounds alone is difficult as there are no pathognomonic signs unique to bacterial or fungal keratitis, meaning diverse organisms can result in similar clinical appearances [4]. Investigations including microscopy and culture are therefore necessary.

Traditionally, microbiological culture has been considered the gold standard for diagnosing microbial keratitis [5,6]. However, this is more relevant in settings where bacterial keratitis is more common compared to fungal keratitis, such as in temperate, high-income settings, as growth on one or more solid media is a diagnostic condition for bacterial keratitis. Fungal growth in a single medium with no associated hyphae visible on microscopy may represent contaminants, whilst conversely, some fungal species are difficult to culture in vitro. As a result, culture has been shown to have a low sensitivity for FK [7] and a variable sensitivity for bacterial keratitis [8]. Microscopy—both in the form of routine “smear” microscopy from corneal scrapings and in vivo confocal microscopy (IVCM)—therefore plays a key role in diagnosing fungal infections [9]. Smear microscopy has the advantage of being fast, inexpensive and accurate, whilst IVCM offers real-time non-invasive diagnosis, with high sensitivity and specificity for diagnosing FK [7,10,11]. A positive finding of fungal hyphae on microscopic examination of corneal epithelial tissue is a highly reliable indicator for ocular fungal infection and should always be treated as significant. However, it may be difficult to interpret the significance of scanty bacteria within corneal smear material, for example, the presence of a very small number of Gram-positive cocci may represent transient flora from the lid margin or conjunctiva, hence the recommendation for supporting cultures [4].

There are several smear microscopy staining techniques available to identify fungal and bacterial organisms. Conventional techniques include Gram stain, potassium hydroxide (KOH) wet mount, Giemsa stain and lactophenol cotton blue [9,12]. These techniques are quick, cheap and easy to perform but their accuracy has been reported to vary considerably, largely due to potential artefacts and misinterpretation. Calcofluor white is inexpensive and quick to prepare but requires a fluorescence microscope [13]. Other techniques include Gomori’s methenamine silver [14], periodic acid-Schiff [15], and fluorescein-conjugated lectins [16], which may be more accurate but are more time-consuming and may require additional equipment and expense.

Although there have been several studies reporting the diagnostic accuracy of IVCM, culture and smear microscopy [7,8,10,11,17,18,19], few have compared these techniques to a composite referent including results of IVCM as opposed to culture alone, particularly within a setting with a high prevalence of FK. In addition, most of these studies have not compared the accuracy of the different microscopic staining techniques. This study aimed to prospectively evaluate the sensitivity of several different smear microscopy stains, IVCM and culture at a tertiary ophthalmic referral centre in Nepal, a setting where there is a high burden of fungal keratitis.

## 2. Materials and Methods

### 2.1. Ethical Statement

This study followed the tenets of the Declaration of Helsinki. It was approved by the London School of Hygiene & Tropical Medicine Ethics Committee (Ref. 14841) and Nepal Health Research Council Ethical Review Board (Ref. 1937). Written informed consent in the local language was obtained before enrolment. If the patient was unable to read, the information was read to them, and they were asked to indicate their consent by application of their thumbprint, which was independently witnessed.

### 2.2. Study Design

This cross-sectional study nested within a prospective cohort study formed part of the triaging assessment used to enrol eligible patients with FK into a randomised controlled trial comparing natamycin 5% to chlorhexidine 0.2%. The full protocol for this study and results have already been published [20,21]. We have previously described the methodology relating to clinical findings, microbiological diagnosis, and in vivo confocal microscopy in our earlier work [22]; we therefore describe these briefly here.

### 2.3. Study Setting and Participants

Patients were prospectively recruited patients at Sagarmatha Choudhary Eye Hospital (SCEH) in Lahan, Nepal between 3rd June 2019 and 9th November 2020. SCEH is a tertiary ophthalmic referral hospital within Province 2 of south-eastern Nepal that serves a population of approximately 5 million people. It is located approximately 18 km from the Indian border, with many patients treated in outpatients being Indian nationals. There are 22 satellite “Eye Care Centres” (ECCs) located within Province 2 that are operated by SCEH and provide routine eye examination and treatment, referring to SCEH for more complex cases and surgery.

Eligible patients were adults (>18 years) with acute MK, defined as having a corneal epithelial defect greater than 1 mm in diameter associated with corneal stromal infiltration, and any/all signs of acute inflammation (conjunctival hyperaemia, anterior chamber inflammatory cells, hypopyon). All eligible patients who consented to participate in the study were included, including those who had received prior antimicrobial treatment.

### 2.4. Clinical Findings

We collected data on demographic details, ophthalmic clinical history and clinical examination findings using a structured case record form, as previously described. [22]. Clinical examination included the best spectacle-corrected visual acuity (BSCVA) in LogMAR and slit-lamp examination, which followed a structured approach: eyelid assessment, corneal ulcer features, anterior chamber characteristics (flare, cells, hypopyon shape, and size), and perforation status.

### 2.5. Microbiological Diagnosis

Laboratory diagnosis was determined using smear microscopy and culture, as previously described [22]. Following the application of preservative-free topical anaesthesia (proxymetacaine), corneal scrape specimens were collected from the base and edge of the ulcer using a slit lamp and 21 G needles. Samples were processed for Gram stain, potassium hydroxide (KOH), and calcofluor white (CFW) preparations as well as direct inoculation on solid culture media (fresh blood agar, chocolate agar, and Sabouraud dextrose agar). Staining using the different techniques was performed in a random order as the microbiologist was not aware of the sequence that the individual scrapes were performed. Media were incubated and read daily at 35–37 °C for bacteria for up to 7 days and at 25–27 °C for up to 21 days for fungi. Organism identification was performed using standard microbiological techniques. We followed a previously described approach for reporting positive microbiological results [4,22]. Culture positivity was used to diagnose BK; smear microscopy alone was not considered to be conclusive evidence. However, if fungal hyphae were visible by smear microscopy, the causative organism was reported as fungal (regardless of culture results).

### 2.6. In Vivo Confocal Microscopy

In vivo confocal microscopy (IVCM) was performed prior to corneal sample collection. IVCM was performed by trained, experienced operators using the HRT III/RCM confocal microscope (Heidelberg Engineering, Dossenheim, Germany) using a previously described technique [10,11,22], with all the images reviewed in real-time and classified into either fungal or amoebic keratitis by one experienced observer. IVCM has been shown to be an accurate tool with good inter-observer agreements in similar settings [10]. IVCM was not used to diagnose bacterial keratitis as the resolution is inadequate to visualise bacteria other than *Nocardia* spp. or to visualise infectious crystalline keratopathy (ICK). The presence of fungal hyphae (defined as highly reflective, branching/bifurcating, well-defined, interlocking structures, measuring 3–10 microns in diameter and not seen in isolation) [23] on IVCM was deemed diagnostic of FK; where there was any uncertainty or the diagnosis was only “possible FK”, the IVCM result would be deemed as negative for the purpose of the analysis.

### 2.7. Disease Definition

Disease definition (fungal, bacterial, amoebic or mixed fungal-bacterial keratitis) was based on positive diagnostic results from culture, smear microscopy and/or IVCM, similar to previous work [7]. Clinical findings or responses to treatment were not used for disease definition to ensure we were assessing the sensitivity of investigations. An overall “composite” diagnosis of definite fungal, bacterial or mixed fungal–bacterial keratitis, or unknown aetiology, was obtained by combining the results of IVCM with cases meeting the microbiological diagnostic criteria described above. This technique of using a composite referent as the “gold standard” has been used previously for these investigations and is appropriate when there is no one acceptable investigation that yields the true number of positive cases, as is the case with the investigations studied here [22]. If all results for each investigation were negative for FK, then the composite diagnosis was negative for FK; if one or more results of the individual investigations were positive for FK, then the composite diagnosis was positive for FK. Missing data were treated as negative in defining the composite diagnosis and excluded for the individual analyses.

### 2.8. Statistical Analysis

Data were analysed in STATA 17 (STATA Corp., College Station, TX, USA). The diagnostic performance of the various tests was evaluated by determining: (1) the sensitivity (including exact binomial confidence intervals) of culture, smear microscopy and IVCM in comparison to a composite reference standard for diagnosing FK, and (2) the sensitivity of the different smear microscopy staining techniques in comparison to the composite reference standard for diagnosing FK. As we are comparing to a composite reference standard, by definition there are no false-positive results, meaning that any calculated specificities would always be 100%. We have therefore chosen not to report specificities to avoid misinterpretation.

## 3. Results

Between 3rd June 2019 and 9th November 2020, 890 patients with suspected MK were assessed at SCEH. Of these, 643 participants consented, with one patient fainting during examination and subsequently withdrawing consent; 642 patients were therefore included in this study. All cases of MK were unilateral (331/643, 51.5% left eye). The clinical characteristics and microbiological aetiology have previously been published [22]. In brief, the median epithelial defect and infiltrate sizes were 2.90 mm and 2.75 mm respectively, whilst *Curvularia* spp. was the most commonly isolated fungal organism (42.8% of cases).

It was not possible to perform all investigations on all patients. For example, due to excessive corneal thinning or not enough material available for smear microscopy and/or culture. Therefore, results from smear microscopy were available in 631/642 cases, IVCM in 638/642 cases and culture for 624/642 cases. Bacterial culture results were not available for 7 cases, whilst fungal culture results were not available for 2 cases.

A causative organism was identified in 565/642 (88%) of cases when all tests were combined (any test positive). The detection method for different types of organisms is shown in Table 1. Most cases that were positive using a composite diagnosis had had all three investigations performed—the few exceptions are mentioned in the text below. Figure 1 illustrates the combination of positive tests for fungal keratitis (including mixed fungal-bacterial infection). We identified 32 cases of monomicrobial bacterial keratitis. The total number of bacterial cases (diagnosed by culture alone), including 65 mixed bacterial-fungal infections, was 97: 15.1% of cases in our study. There was one case of a polymicrobial yeast and bacterial infection, which was detected by culture alone for both the yeast and bacterial infection. There were no cases of *Acanthamoeba* keratitis detected in this study by either IVCM or smear microscopy.

We identified 468 cases of monomicrobial filamentous fungal keratitis: 72.9% of cases in our study. A further 64 mixed fungal–bacterial cases were identified (10%). Including mixed infections, fungal keratitis was diagnosed in 532/642 (82.9%) of all cases of MK in our study. Of these, culture results were not available for 10/532 cases, smear microscopy results were not available for 4/532 cases, and IVCM results were not available for 2/532 cases. Table 1 and Figure 1 show that all three investigations were positive in 355/532 (66.7%) of cases, whilst both smear microscopy and IVCM detected a similar number of FK cases overall (425/464, 91.6% and 421/466, 90.3%, respectively). IVCM detected more cases of FK overall not identified by other modalities (45/532), whilst culture only identified a further 4 cases not detected by IVCM or smear microscopy.

### 3.1. Sensitivity—All Investigations Compared to Composite Referents

The sensitivity of smear microscopy, IVCM and culture for diagnosing FK were compared to the composite reference standard (Table 2). Both IVCM and smear microscopy performed similarly, with sensitivities of approximately 90%, whilst for culture, this was 75% (95% CI 75.4–82.5%).

### 3.2. Sensitivity—Different Smear Microscopy Stains

The sensitivity of the three different smear microscopy stains used for diagnosing FK (KOH, Gram, CFW) was compared to a composite diagnosis reference standard (Table 3). KOH had the highest sensitivity (85.3%, 95% CI 81.9–88.4%), followed by Gram stain (83.2%, 95% CI 79.7–86.4%) and CFW (79.1%, 95% CI 75.4–82.5%). The three different smear microscopy staining techniques were available for 423 cases and compared in Figure 2. All three stains were positive in 336/423 (79.4%) of cases. Gram stain identified 17 cases which the other two techniques did not, KOH 8 cases and CFW 2 cases. KOH identified 396/423 (93.6%) of all cases positive by any microscopy means, Gram stain 385/423 (91.0%), and CFW identified 374/423 (88.4%). There were 34 cases of FK not detected by any of the three compound microscope techniques in the 510 instances where all tests were performed (6.7%).

## 4. Discussion

This study compares the relative sensitivity of three techniques in identifying fungal keratitis in a low-income setting with a high prevalence of fungal keratitis. We have shown in this study that the overall diagnostic yield was high at 88% (565/642) of patients with a clinical diagnosis of MK. Of these, we detected 468 cases of FK, 32 cases of BK and 65 mixed bacterial-fungal infections including one yeast. For FK, whilst all three investigations were positive in two-thirds of cases and all contributed to the overall yield, microscopy (both smear and IVCM) proved the most useful with the highest yield and sensitivities. All investigations contributed additional diagnoses that would have been missed if only two of these three techniques were used. Due to the diagnostic criteria for bacterial keratitis, all cases of bacterial keratitis were diagnosed by culture. Whilst having all three diagnostic techniques increases the overall yield, in settings with limited resources and a high prevalence of fungal keratitis, these results suggest it is worth investing in smear microscopy performed by a suitably trained and experienced microbiologist. However, in these settings, it is still important to remember that culture is required to accurately diagnose bacterial keratitis, and so should be performed at a minimum if no fungal hyphae are visible on microscopy or there is a significant presence of bacteria on Gram stain.

When comparing the three different smear microscopy staining techniques to a composite referent, we found all to fare well with sensitivities of over 79%; KOH performed best with sensitivities of 85.3%, although Gram staining identified the highest number of cases that were not diagnosed by other means (17 compared to 8 detected by KOH and 2 by CFW). Whilst increasing the number of stains performed increases the overall yield, in a resource-limited setting our results suggest that KOH and Gram stain should be the minimum techniques employed. Gram stain has the further advantage of detecting bacterial organisms. Although we did not evaluate Giemsa staining in this study, it is also able to identify bacterial and fungal infections [24], so could be considered if Gram staining is not available. Lactophenol cotton blue is another stain that has been shown to be useful in detecting fungal hyphae [9,12]; however, this was primarily used for fungal identification from subcultured isolates in this laboratory, rather than for initial corneal smear examination, and so we chose not to report these results.

We have previously reported the unusual pattern of fungal organisms responsible in this region, with over half of cases of FK caused by dematiaceous fungi [22], and, including mixed infections, fungal keratitis accounted for 82.9% of all MK cases. Nepal is a country with one of the highest reported incidences of FK globally [25] and is the 29th poorest country in terms of GDP per capita [26]. Most secondary-level hospitals within Nepal have access to basic microbiology facilities, mainly in the form of smear microscopy, with culture facilities found only in tertiary clinical facilities which are mainly located in cities. The same scenario is commonly encountered throughout the LMICs in tropical latitudes, where diagnostic resources are limited or absent and the burden of fungal keratitis is the greatest [3,25]. As a result, cases of microbial keratitis are often treated empirically or based solely on clinical examination, which is known to be inaccurate [27,28]. Anecdotally, there is a reluctance to perform corneal scrapes and send specimens for smear microscopy due to the perceived costs involved and the limited benefit in terms of diagnosing the organism responsible. We feel that this view should be challenged in settings with a high burden of FK: smear microscopy, particularly with KOH, is fast, low-cost, and accurate.

Our finding that smear microscopy gave a high yield with high sensitivities is in contrast to previous work from eastern Nepal conducted between 1998–2001, which found the yield to be low at 48% [29]. This may be due to subsequent improvements in techniques and that this earlier study was conducted in a general hospital using routine microbiology services, as opposed to a dedicated ophthalmic hospital laboratory as in our study. Smear microscopy with KOH gave the highest sensitivity of all smear microscopy techniques for diagnosing fungal keratitis. This finding agrees with results from most other studies from Asia, where the sensitivity of KOH has been demonstrated to range between 80–99.3% when compared to a non-clinical diagnosis-based gold standard (e.g., culture) [9,13,30]. The high sensitivity of smear microscopy for detecting FK in this region is likely in part attributable to the amount of smear material available for analysis; the yield from large ulcers (>2 mm) has been shown to be higher than small ulcers [9]. Patients with FK in our study have large ulcers, with a median ulcer size of 2.55 mm [21]. This is largely due to the late presentation of patients, and inappropriate initial treatment given by pharmacists and traditional healers. Conversely, the yield from smear microscopy in temperate, high-income countries, where patients present earlier in the course of the disease with small infiltrates and bacterial keratitis is far more commonly implicated, is considerably lower [8,31,32], bringing into question its usefulness in the routine diagnosis of microbial keratitis in such settings [33].

Fluorescent staining of fungal material using calcofluor white has previously been shown to be a highly accurate technique, with two studies reporting sensitivities as high as 97% and superior to KOH staining [24,30]. It is believed the high sensitivities are attributable to CFW fluorescence being easy to identify against the background of heterogenous corneal material [34], although it does require an ultraviolet microscope to function, which not all laboratories may have access to. Given the fluorescent nature of the result, it may even be possible for technicians or non-microbiologists to use with limited smear microscopy experience. However, we found the sensitivities of CFW to be lower than previously reported and not as high as KOH. A possible reason for these false negatives may be that the corneal scrape did not contain enough material or there were no hyphae in the portion of the sample taken, or that our “expert” microbiologist was very competent in other staining techniques due to previous experience, meaning CFW, although easy, was not of much additional use. As the processing of the individual slides for different staining techniques was performed in a random order, it is unlikely that the amount of material available for analysis affected these results. Further research comparing the accuracy of non-microbiologists in using CFW is warranted to see if there is a role for it in this setting. It is noteworthy that a previous study from Iran reported the sensitivity of CFW and KOH to be low, at 42% and 28.5%, respectively [35], although the “gold standard” referent in this study was a clinical diagnosis of FK, making comparisons limited.

There have been several previous studies assessing the diagnostic accuracy of IVCM for FK [7,10,17,36]. The two studies from south Asia reported similar sensitivities of 85.7% and 89.2% to what our present study found [10,36]. Of note, the sample size of our present study is larger (642) compared to these (239 and 148). Two studies from the UK, where FK is relatively rare and based on considerably smaller sample sizes (11 and 15), found the sensitivity of IVCM to range between 55.8% and 81.8% and was dependent on the experience of the grader [7,17]. Our study confirms that IVCM is a very useful tool in detecting fungi, particularly in areas with a high prevalence of FK in the hands of experienced graders.

There were several strengths to our study. This was a large, prospective, consecutive-case study, conducted in an area with a high prevalence of FK, that followed a rigid, published protocol and standard operating procedures [20,21]. Most other studies assessing diagnostic accuracy for diagnosing fungal keratitis have used culture as the “gold standard” reference [7,8,10,11,17,18,19]. However, culture is known to have low sensitivity for fungal keratitis [7]. Our study used a composite reference standard rather than culture, which also included IVCM and smear microscopy to detect as many cases as possible.

Limitations to this study are also acknowledged. Firstly, we did not have the facilities for performing PCR on our participants and so we are unable to comment on how this investigation could fit into the overall diagnostic approach for infectious keratitis in our setting. However, the accuracy of PCR for fungal keratitis in a clinical setting has recently been found to be lower than expected [7], whilst PCR remains too expensive for routine use in LMICs. Secondly, when using a composite reference as the “gold standard”, it is not possible to calculate a useful specificity from this data. However, sensitivity could be argued to be of more clinical use in settings where atypical organisms can be encountered and treatment differs considerably between potential diagnoses (e.g., bacterial versus fungal keratitis). Thirdly, our diagnostic criteria for bacterial keratitis relied on positive culture, making any assessment of the accuracy of culture itself impossible, although it should be noted that no diagnosis was available in only 12% of cases overall. Related to this, the number of patients with bacterial keratitis in this study was low. It is possible that bacterial keratitis is more successfully treated than fungal keratitis in secondary eye care settings, leading to more fungal cases presenting at this tertiary level. Additionally, we did not investigate for any correlation between negative diagnostic test results and clinical characteristics. Furthermore, we did not detect any cases of *Acanthamoeba* and only one yeast (polymicrobial mixed bacterial-yeast infection). This may be due to the low prevalence in this setting, the limited diagnostic facilities available, or the lack of experience in detecting *Acanthamoeba* and/or yeasts on IVCM [17]. Finally, we did not have a complete data set for corneal smear slides examined with lactophenol cotton blue-stained smear microscopy because this technique was predominantly used for fungal identification, as is standard practice, and so we did not report the results from this staining technique.

Considering our findings, we propose the following diagnostic approach given in Table 4 for clinicians working in low-resourced settings in tropical and sub-tropical latitudes where fungal keratitis is more prevalent.

## 5. Conclusions

In conclusion, we found that microscopy (both smear and confocal) was the most sensitive tool for identifying fungal keratitis. KOH-wet mount slides appeared to be the most sensitive of the different staining techniques we analysed. Culture remains a useful tool for diagnosing bacterial infections, although in settings where there is a high prevalence of fungal keratitis, a diagnosis can be made based on the presence or absence of fungal hyphae.

## Figures and Tables

**Figure 1 jof-08-00955-f001:**
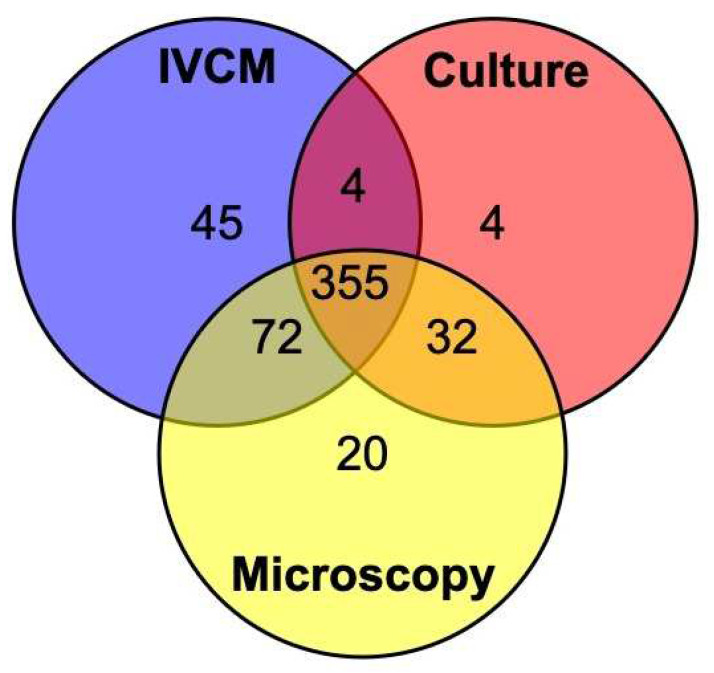
Venn diagram showing the number of cases that were positive for filamentous fungal keratitis (*n* = 532, including mixed bacterial-fungal cases) using culture, smear microscopy, and in vivo confocal microscopy (IVCM). Cases that were positive for more than one test are given within the overlapping areas. Note that cases that were negative for all investigations are not included, and not all patients had all tests performed. Please refer to Table 1 and Table 2 for this information.

**Figure 2 jof-08-00955-f002:**
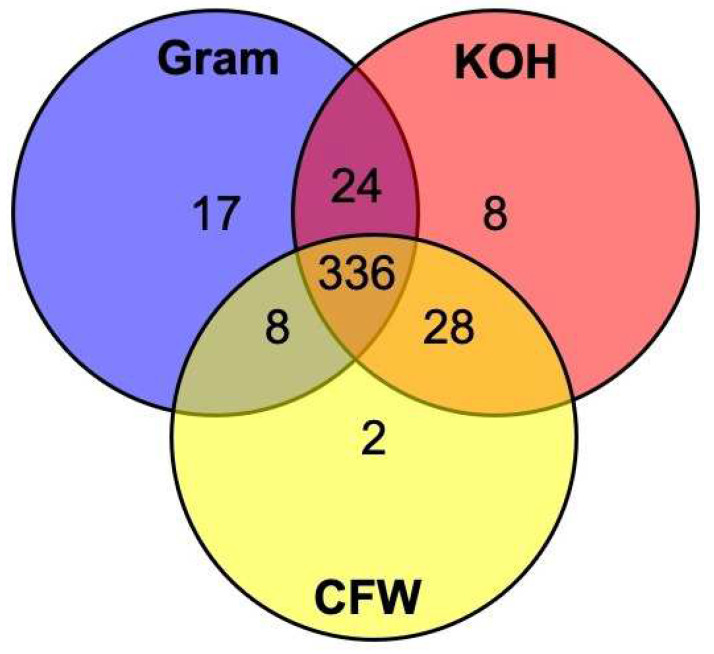
Venn diagram showing the number of cases that were positive for diagnosing filamentous fungal keratitis (including mixed bacterial-fungal cases) by smear microscopy (*n* = 423) using different smear microscopy stains: Gram, potassium hydroxide (KOH), and calcofluor white (CFW). Cases that were positive for more than one stain are given within the overlapping areas. Note that cases that were negative for all investigations are not included, and not all patients had all tests performed. Please refer to Table 3 for this information.

**Table 1 jof-08-00955-t001:** Microbial aetiology for 642 keratitis patients categorised by group of organism and diagnostic techniques, both separately and for the composite diagnosis using all methods combined.

		Diagnostic Methods							
		Composite ^†^		Microscopy		IVCM		Culture	
		*n*	(%)	*n*	(%)	*n*	(%)	*n*	(%)
**Number of subjects tested by each method**		642		631		638		624	
**Positive tests (any organism)**		565	(88.0)	533	(84.5)	476	(74.6)	443	(71.0)
**Positive test by organism type**									
** *Monomicrobial* **	Bacteria ^‡^	32	(5.0)	n/a		n/a		32/32	(100)
Filamentous fungi		468	(72.9)	425/464	(91.6)	421/466	(90.3)	345/458	(75.3)
Yeast		0	(0)	0	(0)	0	(0)	0	(0)
Acanthamoeba ^§^		0	(0)	0	(0)	0	(0)	0	(0)
** *Polymicrobial* ** ^‡^	Bacteria ***WITH***	64	(10.0)	n/a		n/a		64/64	(100)
	filamentous fungi ¶			54/64	(84.4)	55/64	(85.9)	50/64	(78.1)
	Bacteria ***WITH***	1	(0.2)	n/a		n/a		1/1	(100)
	yeast			0/1	(0)	0/1	(0)	1/1	(100)

^†^ The composite reference standard was generated by combining the positive results from microscopy ± IVCM ± culture. ^‡^ Only culture was used to detect bacterial keratitis. ^§^
*Acanthamoeba* were investigated by IVCM and/or smear microscopy only as culture facilities for *Acanthamoeba* were not available. ¶ Bacteria were identified by the results of culture only (microscopy was not used for diagnosis of bacterial keratitis). IVCM, in vivo confocal microscopy. Fungal infection was identified by more than one method; the numbers identified by multiple methods are described in Figure 1.

**Table 2 jof-08-00955-t002:** Sensitivity values for detecting filamentous fungi (*n* = 532) using smear microscopy, in vivo confocal microscopy, and culture compared to a composite diagnosis reference standard (mixed bacterial-fungal infections included). The number of positive and negative test results are shown on the left. The sensitivity values are shown on the right.

Diagnostic Method	Composite Diagnosis Reference Standard ^†^		Totals ^‡^		Value (% CI)	
	Positive	Negative				
**Microscopy**						
Positive	479	0	479	**Sensitivity %**	90.7	(87.9–93.1)
Negative	49	103	152			
**Total**	528	103	631			
**IVCM**						
Positive	476	0	476	**Sensitivity %**	89.8	(86.9–92.3)
Negative	54	108	162			
**Total**	530	108	638			
**Culture**						
Positive	395	0	395	**Sensitivity %**	75.7	(71.8–79.3)
Negative	127	100	227			
**Total**	522	100	622			

IVCM, in vivo confocal microscopy. ^†^ The composite diagnosis reference standard is where an individual tests positive for an organism group (*Acanthamoeba*, bacteria or fungus) in one or more of the three diagnostic investigations. ^‡^ The total number of individuals in the composite diagnosis reference standard differs for each organism group and investigation as not every individual had all three investigations performed. When comparing the tests to the composite reference standard, only the total number of patients who had the particular test in question being performed are included. Please refer to Table 1 for the number of diagnostic tests performed for each organism group in question.

**Table 3 jof-08-00955-t003:** Sensitivity values for detecting filamentous fungal keratitis (*n* = 532) using different smear microscopy stains compared to a composite diagnosis reference standard (mixed infections included). The number of positive and negative test results is shown on the left. The sensitivity values are shown on the right.

Diagnostic Method	Composite Diagnosis Reference Standard ^†^		Totals		Value (% CI)	
	Positive	Negative				
**KOH**						
Positive	413	0	413	**Sensitivity %**	85.3	(81.9–88.4)
Negative	71	95	166			
**Total**	484	95	579			
**Gram stain**						
Positive	417	0	417	**Sensitivity %**	83.2	(79.7–86.4)
Negative	84	59	143			
**Total**	501	59	560			
**Calcofluor white**						
Positive	417	0	417	**Sensitivity %**	79.1	(75.4–82.5)
Negative	110	101	211			
**Total**	527	101	628			

KOH = Potassium hydroxide; CI = Confidence interval. ^†^ Composite diagnosis reference standard is defined as a positive result for at least 1 of the following: culture, smear microscopy or in vivo confocal microscopy.

**Table 4 jof-08-00955-t004:** Proposed diagnostic approach for clinicians working in low-resourced settings in tropical and sub-tropical latitudes where fungal keratitis is more prevalent.

1. **High index of suspicion**. Fungal keratitis should be considered at first presentation, particularly if there are clinical features that are more commonly seen in fungal keratitis such as serrated/feathery margins, satellite lesions, raised slough and pigment [25,37,38].
2. **Perform confocal microscopy (if available).** However, most settings will not have access to IVCM and therefore, clinicians should proceed directly to step 3.
3. **Perform corneal scrapes.** Follow previously published techniques on corneal scraping including how to streak the material on slides and culture media [2,37]. The yield from corneal scrapes increases with increasing ulcer size [9]. Any infiltrates larger than 2 mm must be scraped, and ideally, infiltrates between 1 and 2 mm should also be scraped. However, a negative result in a smaller ulcer may be false–negative and so should be interpreted with caution. Ideally, corneal material should be smeared thinly and evenly onto each microscope slide; if the smear preparation is too thick, this may affect the staining process and will be difficult to interpret. Corneal scraping itself may be of some therapeutic benefit by improving the penetration of topical medications and reducing the infectious burden [2,39]. As a minimum, two slides should be sent (one for Gram stain and one for KOH) for smear microscopy.
4. **Microbiology testing.** We recommend the use of KOH and Gram stain on two separate specimen slides prior to inoculating culture media. Culture is also recommended, especially to diagnose bacterial cases, but facilities may not be available. The use of CFW is a helpful addition if a UV microscope is available as it allows for the hyphae to be clearly visible in relation to the septae. Smear microscopy should be performed as soon as possible after taking the specimen, to enable a diagnosis to be made and treatment to be started before the patient leaves the facility. A strong working partnership with the hospital laboratory is desirable.

## Data Availability

The datasets generated during and/or analysed during the current study will be available upon request from M.J.B. (matthew.burton@lshtm.ac.uk). The full data set will be available with all patient identifiable details removed. Data will be available after formal reporting of the study findings in a peer-reviewed scientific publication. Datasets will only be available to bona fide scientific investigators. Requests should be made to the Chief Investigator in writing detailing the scientific investigator’s background and intended use for the data. Consideration will be given to all proposed analyses, with likely envisaged uses including investigators planning on conducting meta-analyses for example. Patient Information Sheets and consent forms specifically referenced making anonymised data available and this has been approved by the relevant ethic committees.

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
