# Peer review of "Diagnosis of Fungal Keratitis in Low-Income Countries: Evaluation of Smear Microscopy, Culture, and In Vivo Confocal Microscopy in Nepal"

_jof, 2022, doi:10.3390/jof8090955_

Round 1
Reviewer 1 Report
This is a well written research article that addresses diagnosing fungal keratitis using diagnostic tests that are feasible in low-resourced settings. The results and subsequent proposed diagnostic approach will be invaluable for clinicians diagnosing fungal keratitis in such high burden settings. I have a few minor comments:
Methods:
Line 127. Needs full stop inserting after “needles”
Section 2.6: Accuracy of IVCM interpretation could have been improved with agreement between 2 or more observers instead of one. It would be good to know what the interobserver agreement in interpreting scans is in this setting.
Section 2.8: Can a combination of all 3 tests be used to define sensitivity when none is the gold standard? Does this add much else in addition to the individual isolation rates for each individual diagnostic test?
Were patients excluded from the study if they had received prior antimicrobial treatment?
Results:
Table 1: Layout could be improved. Positive tests row – denominator could be removed as is just the total number of subjects in each method in the line above?
Table 2: From looking at table 1 and figure 1, why do the negative totals in table 2 differ from that in figure 1? Please could this be made clearer to the reader.
Table 3: Again, I find the numbers for the composite diagnosis reference standard in this table confusing as they differ from those in figure 2? Please could the authors make this clearer to the reader.
Could the authors include information or where to find the information (reference to paper) regarding the corneal ulcer characteristics (size etc) in the patients studied (mentioned in the discussion section but not the results section) so that the reader can identify how relevant the study findings are to their clinical setting. Was there a correlation between negative diagnostic test results and corneal ulcer characteristics such as previous antimicrobial treatment and corneal ulcer size (this is briefly mentioned in the discussion but if possible it would be good to see if there was an actual correlation from the study results available).
Could the bacterial cases be used to work out specificity of diagnostic methods for fungal keratitis?
Discussion:
Line 295 – remove ? in middle of sentence
Author Response
Thank you very much for taking the time to review this article. Your comments have been very helpful and have helped to improve the manuscript. I have replied to your responses in blue, with the updated manuscript text given in green.
Methods:
Line 127. Needs full stop inserting after “needles”
Thank you. This has been corrected.
Section 2.6: Accuracy of IVCM interpretation could have been improved with agreement between 2 or more observers instead of one. It would be good to know what the interobserver agreement in interpreting scans is in this setting.
Thank you for this comment. I have added a reference to this in the methods section along with the following comment:
IVCM has been shown to be an accurate tool with good inter-observer agreements in similar settings.[10]
Section 2.8: Can a combination of all 3 tests be used to define sensitivity when none is the gold standard? Does this add much else in addition to the individual isolation rates for each individual diagnostic test?
Thank you for this comment. We chose to use a combined referent as the "gold standard" as by doing so we are able to compare each investigation to the known number of true positives; comparing one investigation to another (for example confocal to culture), would give an incorrect number of false positives and would be misleading. This technique of using a composite referent has been used previously when an acceptable gold standard is not available - this has been clarified in the text as detailed below:
This technique of using a composite referent as the “gold standard” has been used previously for these investigations and is appropriate when there is no one acceptable investigation that yields the true number of positive cases, as is the case with the investigations studied here.[22]
Were patients excluded from the study if they had received prior antimicrobial treatment?
Thank you for this comment. Yes, such patients were eligible. I have clarified this in the text as detailed below:
All eligible patients who consented to participate in the study were included, including those who had received prior antimicrobial treatment.
Results:
Table 1: Layout could be improved. Positive tests row – denominator could be removed as is just the total number of subjects in each method in the line above?
Thank you for this useful observation. The Table has been updated as suggested
Table 2: From looking at table 1 and figure 1, why do the negative totals in table 2 differ from that in figure 1? Please could this be made clearer to the reader.
Thank you for this comment. The reason behind this is that Figure 1 only shows positive cases, and does not detail the number of negative cases. I have double checked Figure 1 and the totals of positive cases matches the totals described in Tables 1 and 2. I have made an additional point to clarify this in the footnote:
Note that cases that were negative for all investigations are not included, and not all patients had all tests performed. Please refer to Tables 1 and 2 for this information.
Table 3: Again, I find the numbers for the composite diagnosis reference standard in this table confusing as they differ from those in figure 2? Please could the authors make this clearer to the reader.
Thank you. The reason for this is the same as mentioned above. I have clarified this with the amended footnote:
Note that cases that were negative for all investigations are not included, and not all patients had all tests performed. Please refer to Table 3 for this information.
Could the authors include information or where to find the information (reference to paper) regarding the corneal ulcer characteristics (size etc) in the patients studied (mentioned in the discussion section but not the results section) so that the reader can identify how relevant the study findings are to their clinical setting. Was there a correlation between negative diagnostic test results and corneal ulcer characteristics such as previous antimicrobial treatment and corneal ulcer size (this is briefly mentioned in the discussion but if possible it would be good to see if there was an actual correlation from the study results available).
Thank you for this comment. We have updated this section with the following sentences:
The clinical characteristics and microbiological aetiology have previously been published.[22] In brief, the median epithelial defect and infiltrate sizes were 2.90mm and 2.75mm respectively, whilst Curvularia spp. was the most commonly isolated fungal organism (42.8% of cases).
We did not investigate for any correlation between negative diagnostic test results and clinical characteristics, although this would be interesting to do. This has been added to the discussion section as a potential limitation.
Could the bacterial cases be used to work out specificity of diagnostic methods for fungal keratitis?
Thank you for this question and suggestion. Unfortunately this would not be possible as testing positive for bacterial keratitis does not definitively rule out a fungal infection (and vice-versa).
Discussion:
Line 295 – remove ? in middle of sentence
Thank you, this has been removed.
Reviewer 2 Report
A useful and interesting paper which is well written. The authors outline the relative sensitivities of 3 techniques for diagnosing fungal keratitis in an area and cohort of patients with very high incidence of this disease. This authors highlight the merits of each in the context of available diagnostics in low and middle income countries.
The introduction is well written and relevant
In the methods section it would be useful to consider the order in which samples were collected and processed for the various staining techniques. The authors highlight in the results the relative poor performance of calcofluor white compared to KOH and Gram's stain. This result could be influenced by the amount of material collected and placed on the slide. It is possible if material was always collected in the same order, that the final samples contained less materials than the first. If samples were collected in strict or random order please clarify in the text and consider this in the discussion.
Although the results are complicated, as not every patient had all investigations performed, they are consistently presented. The Venn diagrams are useful in this regard. The authors should review line 191, as Acanthamoeba culture was not performed it could only have been detected by 2 of the three methodologies used.
The discussion need not reiterate the results but in general is well written and the proposed diagnostic approach is useful for clinicians in low and middle income settings
There are one or two typographical errors in the manuscript:
Line 270 the word 'have' is missing 'would have been missed'
Line 295 there is an unnecessary '?' after the word 'incidence?'
Author Response
Thank you very much for taking the time to review this article. Your comments have been very helpful and have helped to improve the manuscript. I have replied to your responses in blue, with the updated manuscript text given in green.
In the methods section it would be useful to consider the order in which samples were collected and processed for the various staining techniques. The authors highlight in the results the relative poor performance of calcofluor white compared to KOH and Gram's stain. This result could be influenced by the amount of material collected and placed on the slide. It is possible if material was always collected in the same order, that the final samples contained less materials than the first. If samples were collected in strict or random order please clarify in the text and consider this in the discussion.
Thank you for this important point. Whilst microbiological slides were taken before culture media were inoculated (in a predefined order), once the three slides reached the microbiologist the order that they were taken was not recorded and therefore the sample processing and staining was performed at random. I have added this in to the methods section, as well as highlighting this in the discussion as follows:
Methods: Staining using the different techniques was performed in a random order as the microbiologist was not aware of the sequence that the individual scrapes were performed.
Discussion: As the processing of the individual slides for different staining techniques was performed in a random order, it is unlikely that the amount of material available for analysis affected these results.
Although the results are complicated, as not every patient had all investigations performed, they are consistently presented. The Venn diagrams are useful in this regard. The authors should review line 191, as Acanthamoeba culture was not performed it could only have been detected by 2 of the three methodologies used.
Thank you for this comment. We have clarified the statement on Acanthamoeba as follows:
There were no cases of Acanthamoeba keratitis detected in this study by either IVCM or smear microscopy.
There are one or two typographical errors in the manuscript:
Line 270 the word 'have' is missing 'would have been missed'
Line 295 there is an unnecessary '?' after the word 'incidence?'
Thank you for highlighting these. These have now been corrected.